# Bone Regeneration in Small and Large Segmental Bone Defect Models after Radiotherapy Using Injectable Polymer-Based Biodegradable Materials Containing Strontium-Doped Hydroxyapatite Particles

**DOI:** 10.3390/ijms24065429

**Published:** 2023-03-12

**Authors:** Camille Ehret, Rachida Aid, Bruno Paiva Dos Santos, Sylvie Rey, Didier Letourneur, Joëlle Amédée Vilamitjana, Erwan de Mones

**Affiliations:** 1INSERM U1026, Tissue Bioengineering, University of Bordeaux, 33076 Bordeaux, France; c.ehret@siltiss.fr (C.E.); brnpaivas@gmail.com (B.P.D.S.); sylvie.rey@inserm.fr (S.R.); erwan.de-mones-del-pujol@chu-bordeaux.fr (E.d.M.); 2SILTISS SA, Zac de la Nau, 19240 Saint-Viance, France; 3Université Paris Cité, INSERM U1148, LVTS, X Bichat Hospital, 75018 Paris, France; rachida.aid@inserm.fr; 4Université Paris Cité, INSERM UMS-34, FRIM, X Bichat School of Medicine, 75018 Paris, France; 5CHU Bordeaux, Department of Oto-Rhino-Laryngology, 33076 Bordeaux, France

**Keywords:** irradiated bone tissue, strontium-doped composite polymer, vascularization, bone reconstruction

## Abstract

The reconstruction of bones following tumor excision and radiotherapy remains a challenge. Our previous study, performed using polysaccharide-based microbeads that contain hydroxyapatite, found that these have osteoconductivity and osteoinductive properties. New formulations of composite microbeads containing HA particles doped with strontium (Sr) at 8 or 50% were developed to improve their biological performance and were evaluated in ectopic sites. In the current research, we characterized the materials by phase-contrast microscopy, laser dynamic scattering particle size-measurements and phosphorus content, before their implantation into two different preclinical bone defect models in rats: the femoral condyle and the segmental bone. Eight weeks after the implantation in the femoral condyle, the histology and immunohistochemistry analyses showed that Sr-doped matrices at both 8% and 50% stimulate bone formation and vascularization. A more complex preclinical model of the irradiation procedure was then developed in rats within a critical-size bone segmental defect. In the non-irradiated sites, no significant differences between the non-doped and Sr-doped microbeads were observed in the bone regeneration. Interestingly, the Sr-doped microbeads at the 8% level of substitution outperformed the vascularization process by increasing new vessel formation in the irradiated sites. These results showed that the inclusion of strontium in the matrix-stimulated vascularization in a critical-size model of bone tissue regeneration after irradiation.

## 1. Introduction

Bone tissue regeneration following bone tumour excision remains a challenge for large and complex bone defects [1]. Indeed, the use of an autologous bone graft constitutes the standard strategy for bone regeneration [2]. However, this approach has some limitations; two surgical procedures are required, and the infection risks, morbidity at the donor site, and reduced volume have been identified. With the use of allografts, additional risks concern the transmission of infectious diseases and immunological reactions. Vascularized fibular or iliac bone grafts offer significant advantages, but also significant disadvantages, which include the facts that it is time-consuming, technically difficult, and comes with no guarantee of satisfactory functional results [3].

Thereafter, tissue engineering and the development of osteoconductive and osteoinductive matrices can provide innovative solutions for bone reconstruction [2,3,4]. However, the bone tissue engineering strategies used today are still a long way from being applied in oncology as a form of immediate post-implantation irradiation. Bone tumor resection can result in a large bone defect, and frequently, the irradiation procedure decreased the osteoinductive and pro-angiogenic potential of the potential grafts used for bone reconstruction [5,6,7,8]. Within this context, large bone segmental defects cannot completely heal with the use of only one-step autologous spongious bone grafts. These bone grafts resorb too quickly [9]. To avoid this issue, the use of an anatomical barrier between the graft and the surrounding non-osseous site can limit this resorption [10]. Moreover, this barrier can partially preserve the biological properties of the grafts.

Among the other alternatives, Masquelet described the procedure of the induced membrane surgical procedure technique for long bone segmental defect reconstruction [11,12]. This strategy can be used in patients for bone reconstruction after cancer treatment. Using this technique, radiotherapy can affect the two-step surgical procedures: firstly, the biological properties of the induced membrane produced from polymethyl meythacrylate (PMMA) [13], i.e., pro-angiogenic properties; and secondly, the ability of the autografts loaded within the membrane to rebuild an irradiated avascular site. Regarding the first surgical step, our previous works showed that induced membranes produced using polymethyl methyacrylate (PMMA) maintained their histological and biochemical properties after the external beam radiotherapy (EBRT) procedures [14]. For the second step, several alternatives to autografts are currently proposed, such as the supplementation of bone substitutes with bone marrow suspension or with bone morphogenetic proteins (BMPs) [15,16]. However, despite there being few publications describing the effective reconstruction of irradiated wounds using rhBMP2 [17], supplementation with BMPs also remains contraindicated in patients who have an active malignancy or who are undergoing treatment for malignancy. Others authors have reconstructed large femoral segmental critical-size defects in rabbits with a biomaterial that has a resorbable collagen membrane, MBCP^®^, combined with post-radiation autologous total bone marrow grafting [18]. The total bone marrow associated with biphasic calcium phosphate significantly enhanced bone formation in irradiated bone, probably by only keeping the soluble factors, such as growth factors and cytokines [19]. Whatever the procedures, the revascularization of the biomaterials remains the main limitation in efficient bone reconstruction, especially in irradiated sites [3]. 

Here, we evaluate a cell-free and growth factor-free strategy for irradiated bone reconstruction. In addition to the calcium phosphate-based materials described in numerous works, the polysaccharide-based microbeads that contain hydroxy-apatite that were used in the present work have already been extensively described by our teams for bone tissue engineering applications [20,21,22,23,24,25,26]. Our previous publications demonstrated the potential of these natural, biodegradable, cell-free and growth factor-free polysaccharide-based matrices that are available as ready-to-use sterile injectable biomaterials for craniofacial and dental applications [21,24]. We have also demonstrated that the initial radiolucent property of these matrices at the beginning of implantation is of particular interest from a clinical point of view, as bone formation inside a grafted material can be observed using a conventional X-ray analysis, unlike to conventional calcium phosphate-based materials or allogenic bone substitutes. In addition, our previous works demonstrate that our polysaccharide-based matrix supplemented with HA is osteoinductive and osteoconductive [21]. These properties were evidenced in ectopic sites (subcutaneously) and in orthotopic sites (femoral condyle in rats, sinus lift model in sheep) [22,24]. Once implanted subcutaneously in rats, or intramuscularly in sheep, these composite matrices induced mineralized and osteoid tissues in the two ectopic models.

Thereafter, these polysaccharides-based matrices were doped with two ratios of strontium (Sr^2+^) at 8% or 50%, incorporated in the HA components (8Sr-HA and 50Sr-HA) [23], to improve their performance for bone repair. In vitro studies showed that these Sr-doped matrices supported the osteoblastic differentiation of human mesenchymal stem cells and stimulated the expression of osteopontin, a late osteoblastic marker involved in the mineralization events. In vivo, these Sr-doped matrices stimulate the osteoid tissue, as well as the blood vessels formation after two weeks of subcutaneous implantation [23].

Other recent in vitro studies that focused on Sr-substituted calcium phosphates in combination with polymers (i.e., chitosan) proved the increased osteogenic differentiation of human mesenchymal stem cells compared to materials containing non-substituted calcium phosphates [27]. More recently, the incorporation of strontium-containing HA in a 3D printed polycaprolactone (PCL) scaffold has shown an increase in osteogenic-related gene expression in bone marrow stromal cells in vitro, compared to PCL or to PCL-HA scaffolds [28]. In the same way, Chandran et al. demonstrated in vitro an increase in the alkaline phosphatase activity in sheep adipose tissue-derived mesenchymal stem cells, cultured on strontium–calcium phosphate scaffolds, compared to calcium phosphate scaffolds deprived of strontium [29]. These authors evidenced that the implantation of these calcium-doped phosphate scaffolds in a sheep osteoporotic model facilitates osteogenesis and osteointegration [29].

Based on the literature and our previous findings in a polysaccharide-based matrix supplemented with HA and doped with Sr, our objectives in the presented work were to evaluate the ability of these Sr-doped matrices to regenerate two different bone defects. Two new challenges were also defined, i.e., an evaluation in a critical-size defect of load-bearing bone using a polysaccharide matrix of initially low mechanical properties, and in a segmental bone defect after an irradiation procedure. Here, we qualitatively and quantitatively analyzed the performance of these innovative Sr-doped matrices in supporting bone formation, as well as vascularization in these two models: (i) in a non-critical-size bone defect model (rat femoral condyle) [22] and (ii) in critical-size and irradiated bone sites (rat segmental bone defect), one of the most complex clinical scenarios for orthopedic and oral surgery. A new preclinical model of irradiation in rats was developed here (Figure 1) to be close to the tumor bone resection clinical situation, and then, we evaluated the efficacy of the Sr-doped matrices in promoting vascularization and bone formation. 

## 2. Results

### 2.1. Characterization of the Microbeads of Matrix-HA Doped with Strontium

Three formulations of composite beads (Matrix-HA, Matrix-8Sr-HA, and Matrix-50Sr-HA) were prepared with the same concentration of polysaccharides and hydroxyapatite, doped or not with strontium. The composite beads were hydrated in PBS and observed by phase-contrast microscopy, with all formulations showing identical regular surfaces and spherical shapes (Figure 2A). 

Laser dynamic scattering was used to evaluate the size distribution of the hydroxyapatite +/− strontium particles, and of the beads with HA +/− strontium (Figure 2B). Size analysis revealed that the median size (50%) of HA particles (doped or not with strontium) was 31 ± 9 µm diameter (range: 7± 1.5 µm–103 ± 40 µm). The median size (50%) of the polysaccharide beads was 427 ± 40 µm in diameter (range: 218 ± 52 µm–698 ± 89 µm). 

The phosphorus content of the composite beads is due to the phosphorus from hydroxyapatite, and also to the phosphorus from the crosslinking. The phosphorus contents were similar for all formulations, with 531 ± 10 µmol/g of dried composite beads for Matrix-HA, 495 ± 18 µmol/g for Matrix-8Sr-HA and 529 ± 17 µmol/g for Matrix-50Sr-HA (Figure 2C). 

### 2.2. In Vivo Evaluation of the Matrix-HA Supplemented with Strontium Implanted in the Femoral Condyle Bone Defects

The tissues formed after 4 (W4) and 8 weeks (W8) of implantation of the microbeads in the femoral condyle bone defects were first analyzed by micro-CT. The results in Figure 3A showed that microbeads of the three groups (Matrix-HA, Matrix-8Sr-HA, Matrix-50Sr-HA) were still present within the condyle defects, and that mineralization occurred in the periphery and within the microbeads, mainly after 8 weeks of implantation. Quantitative analyses of these images (Figure 3B) using the mineralization values (ratio of mineralized volume over total volume—MV/TV) showed an increase in MV/TV in terms of the time of implantation for Matrix-HA and Matrix-8Sr-HA. A significant value of the MV/TV was already observed 4 weeks after the implantation of matrices containing 50% of strontium substitution (Matrix-50Sr-HA), compared to the matrices deprived of strontium (Matrix-HA) at the same time point (W4). No significant difference between the two time points (W4 and W8) for Matrix-50Sr-HA was observed.

Masson’s trichrome staining of the newly formed tissue (Figure 4A) and the quantification of the osteoid tissue (Figure 4B) confirmed, at 8 weeks (W8), that the supplementation of strontium in the matrices, whatever the dose (Matrix-8Sr-HA or Matrix-50Sr-HA) significantly stimulated the osteoid tissue formation compared to Matrix-HA. This effect was not yet observed at 4 weeks, whatever the dose of strontium. 

The formation of vessels was also analyzed by vWF immunostaining performed in the three groups (Figure 5A). Whatever the time points (W4 and W8), both Matrix-8Sr-HA and Matrix-50Sr-HA groups significantly increased the number of vessels/mm², unlike the Matrix-HA that was devoid of strontium (Figure 5B).

### 2.3. In Vivo Evaluation of the Matrix-HA Supplemented with Strontium Implanted in a Critical-Size Segmental Bone Defect in Rat after Irradiation

The radiation procedure was performed as described in the Materials and Methods section on the segmental bone defects treated with the three groups of matrices (Matrix-HA, Matrix-8Sr-HA, and Matrix-50Sr-HA) and was summarized in Figure 1. For non-irradiated and irradiated animals, bone explants were analyzed 12 weeks after the microbeads implantation and analyzed by micro-CT, histology, and immuno-histochemistry.

The external radiation was completed without major complications for all groups during radiation procedures. Infection after surgical implantation procedures was not observed. With or without radiation, micro-CT images at 12 weeks revealed that the mineralization occurred in the bone cavity, inside and surrounding the microbead (Figure 6A). This mineralization was observed mainly at the periphery, surrounding the bone defect, decreasing gradually toward the center, whatever the groups of the implanted microbeads (Figure 6A). Quantitative analysis revealed no significant difference between the three groups of matrices (Matrix-HA, Matrix-8Sr-HA, and Matrix-50Sr-HA) in both the non-irradiated (Figure 6B) and irradiated sites (Figure 6C). We noticed that the MV/TV values were lower in the irradiated sites than in the non-irradiated sites (Figure 6B,C).

Histologically, the new bone formation (osteoid and lamellar bone, including the lacunae of osteocytes) was observed surrounding the microbeads in all groups of bone explants in direct contact with the material (Figure 7A). This effect was observed in the absence of or after an irradiation procedure. The bone ingrowth seems to be homogeneous within the bone cavity. High magnifications of different areas of the bone defect, for both groups (non-irradiated versus irradiated site), revealed that osteoid tissue was present in the center of the bone defect (Figure 7A), as well as in the union area with the host tissue (Figure 7B). For both irradiated and non-irradiated groups, the beads of the matrix were well-integrated in the union area (Figure 7B), whatever the ratio of strontium substitution. 

The quantification of osteoid tissue in both conditions (non-irradiated versus irradiated site) (Figure 7C,D) and for the three groups revealed that Sr substitution only influenced the amount of osteoid tissue in the non-irradiated groups (Figure 7C). A significant increase in osteoid tissue was observed for Matrix-50Sr-HA compared to Matrix-HA. For the irradiated group, Sr substitution did not significantly influence the amount of osteoid tissue (Figure 7D).

The formation of vessels was followed by vWF in the three groups of matrices, and for both groups (non-irradiated versus irradiated site) (Figure 8A). Quantitative analysis revealed that the radiation procedure has a negative impact of the number of vessels/mm^2^ formed in the newly formed tissue compared to the non-irradiated groups. In the non-irradiated group (Figure 8B), the Sr substitution did not significantly modulate the number of vessels/mm^2^, whatever the ratio of substitution (8Sr or 50Sr). In contrast to Matrix-HA and Matrix-50Sr-HA, Matrix-8Sr-HA significantly stimulated the number of vessels/mm^2^ in the newly formed tissue of the irradiated group (Figure 8C).

## 3. Discussion

Here, we first aimed to measure the impact of the Sr substitution on polysaccharide-based matrices supplemented with HA on bone formation in a non-critical-size bone defect in rats (femoral condyle) and in a non-irradiated critical-size bone defect (femoral segmental defect). Using the microbeads of Matrix-HA doped with Sr, we studied their effect on cellular and tissue invasion within these bone defects. Secondly, we evaluated the potential of these microbeads—devoid of growth factors or autologous mesenchymal stem cells—to repair bone defects in the most complex clinical context: after the radiation procedure that often follows a bone tumor excision. 

Indeed, the method to restore bone defects after ablative tumor surgery and radiotherapy remains a clinical challenge. The hypocellularity, hypovascularity, and hypoxia conditions induced by the irradiation inhibits bone healing. The apoptosis of many cells in the irradiated site occurs after radiotherapy, including osteoprogenitor cell osteocytes and vascular endothelial cells, leading to the progressive fibrosis of this host avascular tissue that is not conductive to bone reconstruction. In such a complex physiological situation, our approach was to develop a cell-free and growth factor-free scaffold for bone reconstruction. 

Our previous works demonstrated that a polysaccharide-based matrix supplemented with HA is osteoinductive and osteoconductive [21]. These properties were evidenced in ectopic sites (subcutaneously) and in orthotopic sites (femoral condyle in rats, sinus lift model in sheep) [22,24], but not in a critical-size bone defect such as the segmental bone defect models developed here. Since these polysaccharide-based matrices exhibit low initial mechanical properties, we developed strontium substituted hydroxyapatite (Sr-HA) polysaccharide-based microbeads for applications in defects of load-bearing bone aiming to accelerate mineralization and bone formation. Adding Sr into bone graft substitutes, mainly in calcium phosphate materials, has already triggered great interest in the treatment of critical-sized bone defects [30]. 

In this research, microbeads of Matrix-HA, doped or not with strontium, were produced and characterized before their implantation in bone defects. In our previous work, these matrices, doped or not with strontium, were characterized by environmental scanning electron microscopy (ESEM) using inductively coupled plasma optical emission spectrometry (ICP-OES) for the determination of the strontium (Sr) and calcium (Ca) content [23]. With increasing Sr substitution in the HA particles (8% and 50% of molar substitution), ICP-OES showed an increasing amount of Sr in the HA samples from 2 ± 1 μg/mg of powder (8Sr-HA) to 9.7 ± 3.8 μg/mg of powder (50Sr-HA). Here, these microbeads, doped or not with strontium, were analyzed by phase-contrast microscopy, laser dynamic scattering particle size measurement, and the colorimetric method for their phosphorus content. The composite beads used in this study were the result of an emulsion synthesis method and the use of a sieving process. The three formulations presented a spherical shape with a regular surface. The composite beads in PBS had a median size of 430 μm. In the three formulations, the phosphorus content, due to the crosslinking and presence of hydroxyapatite, was similar (Figure 2). 

Regarding the Sr^2+^ release from these microbeads, several in vitro studies reported Sr^2+^ release from Sr-doped biomaterials using a similar substitution ratio as our Matrix-8Sr-HA [31,32,33,34]. In most of these studies, the in vitro Sr^2+^ release was determined using ICP-OES, in water, in Hanks’ balanced salt solution, or in simulated body fluid, as well as in static and simulated dynamic conditions. However, in vitro models are highly dependent on the immersing media [35] (water versus simulated body fluid, flow, and pH) and did not take into account a possible exchangeable pool of strontium linked to proteins, and no cellular event is present in vitro. It is likely that bone remodeling and the in vivo environment accelerate the Sr^2+^ release by mechanisms such as biomaterial resorption by cells such osteoclasts, and numerous enzymatic activities found in the injured area. 

In our work, although the in vitro Sr^2+^ release from these Matrix-8Sr-HA and Matrix-50Sr-HA was not determined, Sr^2+^ release and its effect on the osteoblastic differentiation of human mesenchymal stem cells were previously demonstrated [23]. Matrix-8Sr-HA and Matrix-50Sr-HA both activated the expression of one late osteoblastic marker involved in the mineralization process, i.e., osteopontin [23], unlike the Matrix-HA devoid of strontium. In addition, these Sr-doped matrices impacted in vivo osteoid tissue and blood vessels formation. Future in vivo studies are thus required to carefully decipher the mechanisms of dissolution of our Sr-HA biomaterials.

Here, our objectives were to test the ability of these doped matrices to regenerate two different bone defects, a non-critical-size bone defect, i.e., the femoral condyle bone defect, and a critical-size defect of load-bearing bone, i.e., the segmental bone defects in rats after an irradiation procedure.

The use of these matrices as microbeads seems to be relevant in these two preclinical models, and other studies have highlighted the fact that microbeads are more suitable than massive blocks for tissue invasion and bone tissue engineering [29,36]. Indeed, with Masson’s trichrome staining for both models (femoral condyle and segmental bone defect), the colonization of osteoid tissue through and around the microbeads was clearly visible in the periphery and in the centre of the bone defects (Figure 4 and Figure 7A,B). In addition, the presence of vascularisation in the middle of the femoral condyle bone defect (Figure 5) could be attributed, in part, to the effect of strontium within the matrices, and, in another part, to the presence of beads that facilitate vessel migration through the defects. 

Regarding the ability of these microbeads, doped or not with Sr, to regenerate bone defects in the femoral condyle of rats, quantitative results demonstrate that the presence of strontium within the matrix modulates the mineralization of the tissue, and accelerates the kinetics of mineralization (Figure 3B) in Matrix-50Sr-HA, unlike non-doped microbeads. As previously described [23], it is important to note that this material, doped or not with Sr, showed radiolucent properties before implantation, and that the increase in the signal is only due to the mineralization of the newly formed tissue. We also evidenced here that both ratios of Sr substitution (Matrix-8Sr-HA and Matrix-50Sr-HA) stimulate the formation of vessels within the newly formed tissue (Figure 5). The optimal ratio of Sr substitution for increasing the formation of osteoid bone tissue (Figure 3 and Figure 4) require further experiments. 

Then, we implanted three different groups of microbeads in a critical-sized bone defect. We applied the Masquelet technique [12] to create a segmental femoral defect in rat model. First, we studied the effect of the microbeads, doped with strontium at different ratios (Matrix-8Sr-HA and Matrix-50Sr-HA), on the tissue mineralization and vascularization. As the Masquelet technique is often used in the context of bone resection following tumor ablation, radiotherapy is currently required to eradicate residual tumoral cells. We thus studied the ability of these Sr-doped matrices to regenerate bone tissue after the external beam radiotherapy (EBRT) of the femur.

In absence of EBRT, we hypothesized that the strontium-doping of the osteoinductive and osteoconductive polysaccharides-based matrices could increase vascularization and, then, bone formation. The results presented in Figure 6A,B and Figure 7C reveal that, 12 weeks after implantation, mineralization occurs within the defect. Osteoid and bone tissue formed around the beads of matrices, as well as lamellar bone with the presence of osteocytes, whatever the area of analysis (Figure 7A,B), including in the center of bone defect. Histological studies confirmed that microbeads are well-integrated in the interface of the bone defect and surrounding tissue (Figure 7B), whatever the group of microbeads (Matrix-HA, Matrix-8Sr-HA, and Matrix-50Sr-HA). Regarding the time points of implantation, bone defects were analyzed 12 weeks post-implantation, as described by other authors mainly using similar experimental models [37]. However, in most of these experiments, scaffolds were associated with autologous bone mesenchymal stem cells, adipose-derived stem cells, autologous bone marrow cells, and/or deliver growth factors such as VEGF or BMPs [38,39,40]. 

In these studies, bone reconstruction was observed at 8 to 16 weeks post-operatively, despite the combination of the scaffolds with osteoinductive or angiogenic elements (cells or growth factors). In our work, quantitative results showed that these microbeads—devoid of cells and growth factors—were able to induce bone tissue regeneration homogenously in critical-sized bone defects in rat after 12 weeks. 

In the clinical context, these large bone defects on load-bearing bones arise from trauma and pathology and following tumor excision. Therefore, a method to restore large bone defects after ablative tumor surgery and radiotherapy remains a clinical challenge [1,41]. The advantages and disadvantages of the surgical alternative for autografts or allografts to reconstruct massive bone defects after tumor excision have been reported, and most of the strategies are focused on the use of vascularized grafts (fibular, iliac bone graft, etc.) [3] and/or bone lengthening with external fixation. The other procedures consisted of the combination of bone substitutes with mesenchymal stem cells, associated or not with endothelial cells [42], or even bone substitutes combined with growth factors (BMPs) [17], while these BMPs can exert adverse effects in an oncological context. More recently, miR-34 was used in order to contribute to bone regeneration in irradiated bone defects, by enhancing osteoblastic differentiation of mesenchymal stromal cells in rats [43]. However, if these growth factor/molecules and cell delivery have shown great therapeutic potential for bone regeneration after radiotherapy, no data were reported showing the benefit of a scaffold deprived of reparative cells and growth factors for the bone reconstruction of an irradiated area, as observed here. 

For the radiation procedure used in this work, our group has already evaluated the mode of irradiation and its impact on the spacer of PMMA and the induced membrane produced according to the Masquelet method [14]. One of the important events in radiotherapy in animal models is the choice of dosage and fractionation mode [5]. Single 2 Gray (Gy) irradiation has often been used in animal models [6]. In addition, the choice of irradiation site can also affect the radiation effects, while total body irradiation may cause an inflammatory response. Here, a dose of 55 Gy was delivered over 5 weeks and 5 weeks before the second surgical step, when microbeads of Matrix-HA, Matrix-8Sr-HA, or Matrix-50Sr-HA were implanted post-radiation procedure. It could be interesting to assess the nature of the irradiated tissue before the implantation of the groups of matrices in order to evaluate the vascular damage induced, in particular, by the EBRT. 

Following the EBRT treatment, no severe side effects were observed in the group of irradiated animals. Only a slight alopecia was detected and transient, since it was not visible anymore at the time of the euthanasia, 12 weeks after the end of the EBRT. We compared the formation of the mineralized tissue 12 weeks after implantation (Figure 6), in addition to the osteoid formation (Figure 7), and the vascularization (Figure 8) in the segmental bone defects of the two groups of rats i.e., non-irradiated versus irradiated rats, treated with the three groups of microbeads (Matrix-HA, Matrix-8Sr-HA, or Matrix-50Sr-HA). Micro-CT analysis demonstrated that the MV/TV ratio was comparable between the two groups of rats, irradiated or not, whatever the microbeads implanted (Matrix-HA, Matrix-8Sr-HA, or Matrix-50Sr-HA). This suggests that the radiation procedure did not affect the degree of mineralization induced by the microbeads of Matrix-HA. Within the irradiated group, the Sr substitution, whatever the ratio of Sr, did not significantly modify the MV/TV ratio (Figure 6C). Interestingly, the accuracy of the micro-CT system appeared to be sufficient for comparative measurements of the mineralization density, as well as the distribution of the mineralization within the defects, indicated by the structure opacity. Higher mineralization density of the microbeads close to the bone ends of the defects were observed, whatever the microbeads implanted (Figure 6A). This observation of the mineralized tissue was histologically confirmed by the presence of well-engineered integrated microbeads, in both bone ends of the defects, all surrounded by a dense lamellar bone tissue for all conditions (Figure 7B, non-irradiated sites and irradiated sites). As discussed previously, these analyses were all performed 12 weeks post-implantation and longer-term studies could identify possible differences between the groups. Histological analyses (Figure 7) showed significant differences between the Matrix-HA and Matrix-50Sr-HA groups in non-irradiated group of rats, an effect that we did not observe in the irradiated rat groups at the same time point. Finally, the quantitative analysis of the neo-vascularization in the irradiated sites evidenced a positive effect of Matrix-8Sr-HA, unlike the microbeads of Matrix-50Sr-HA and non-doped microbeads (Matrix-HA). We also noticed that the number of vessels/mm^2^ remained lower in the irradiated sites compared to the non-irradiated site, whatever the group of microbeads implanted. As expected, this result confirms that the radiotherapy compromises the vascularization of the tissue and, thereafter, optimal bone reconstruction. The very low level of vascularization of the irradiated sites compared to the non-irradiated sites could explain the impact of the Matrix-8Sr-HA only in the irradiated sites and not in the non-irradiated sites, where the number of vessels was around three times higher (Figure 8), and for which the effect could be less detectable. Then, as observed here for the femoral condyle bone defect model, Sr substitution and, mainly, Matrix-8Sr-HA can modulate the vascular damage induced by radiotherapy. However, it is not yet possible to define the optimal ratio of strontium in the microbeads to achieve the optimal ratio for both vascularization and bone formation in this preclinical scenario. 

## 4. Materials and Methods

### 4.1. Synthesis of Hydroxyapatite Particles

Hydroxyapatite (HA) particles were synthesized by wet chemical precipitation [44] with some modifications. Briefly, HA particles were synthesized using 50 mL of 1.08 M Ca(NO_3_)_2_, 4H_2_O solution, and 50 mL of 0.65 M (NH_4_)_2_HPO_4_ solution, heated at 90 °C. The pH of the solution was adjusted to 10 with NH_4_OH, added dropwise under stirring. The precipitate was maintained for 5 h at 90 °C under stirring, and then washed 4 times with CO^2^-free distilled water. Strontium-substituted hydroxyapatite (Sr-HA) was obtained following the same procedure, by adding Sr^2+^ions into the Ca^2+^ solution, before adjusting the pH to 10 with NH_4_OH, in accordance with our previous work [23]. For this purpose, the appropriate amounts of Sr(NO_3_)_2_ and Ca(NO_3_)_2_·4H_2_O were dissolved in order to obtain 8% (8Sr-HA) or 50% (50Sr-HA) (molar ratio) of Ca^2+^ substituted by Sr^2+^, respectively [23].

### 4.2. Synthesis of the Composite Beads of Polysaccharide-Based Matrices Doped or Not with Strontium

Composite beads were produced as previously described [21,22,24]. Three formulations of beads (Matrix-HA, Matrix-8Sr-HA, and Matrix-50Sr-HA) were prepared with the same concentrations of polysaccharides and hydroxyapatite that were doped or not with strontium. Briefly, pullulan and dextran 75:25 (Pullulan, Mw 200,000, Hayashibara Co. LTD Okayama, Japan; Dextran, Mw 500,000, Pharmacosmos A/S, Holbaek, Denmark) were dissolved in hydroxyapatite suspensions (5% *w*/*v*), doped or not with strontium. The cross-linking of the polysaccharides was carried out using sodium trimetaphosphate (STMP, 2.5% (*w*/*w*), Sigma-Aldrich, Saint Quentin Fallavier, France) under alkaline conditions, as previously reported [20,25] To form the microbeads, the solution was dispersed in oil under mechanical stirring. After freeze drying, dried spherical beads were obtained. The biological and physico-chemical properties of the materials (Phosphorus, HA and Sr content, size distribution of the beads, microscopic surface observations) have already been described in previous works [23]. Freeze-dried beads of Matrix-HA, Matrix 8Sr-HA, and Matrix 50Sr-HA were then packaged in 1 mL or in 5 mL syringes, sterilized by gamma irradiation at a dose of 25 – 35 kGy (Gammacell 3000 Elan, MDS Nordion, Ottawa, Canada) and stored at room temperature until use. 

### 4.3. Morphology 

The freeze-dried composite beads were hydrated in PBS 0.1M pH 7.4 and digital photos were taken using phase-contrast microscopy (OLYMPUS BHA) with a ×10 camera lens. Hydroxyapatite and composite beads size distribution were determined using the Mastersizer 3000 Laser (Malvern Instruments, Orsay, France) after hydration in PBS 0.1M pH 7.4 or in water. Six measurements were made for each formulation.

### 4.4. Phosphorus Quantification

Composite beads were obtained by the cross-linking of the polysaccharides using sodium trimetaphosphate (STMP). The phosphorus content of the beads was quantified to confirm that phosphate bridges between polysaccharides chains were well established in the three formulations. The phosphorus content of the composite beads was determined according to the colorimetric method. About 20 mg of the freeze-dried materials was incubated in 10% HNO_3_ at 105 °C until gel dissolution was complete. Ammonium metavanadate and ammonium heptamolybdate were then added to the dissolved hydrogel solution and optical density was determined at 405 nm by spectrophotometry (Infinite M200 Pro, TECAN^®^, Männedorf, Switzerland). The phosphorus content was determined using a calibration range with a phosphoric acid (H_3_PO_4_) solution. Four samples were tested, and the results are expressed as mean values ± standard deviation (SD).

### 4.5. Bone Defect in the Femoral Condyle in Rat

The condyle bone defect model in rats was used as a first model to evaluate the impact of strontium on bone formation and revascularization. Female rats (250–300 g) were used (Charles River Laboratories, Lyon, France). All the procedures for rat handling were based on the principles of Laboratory Animal Care formulated by the National Society for Medical Research and approved by the Animal Care and Experiment Committee of University of Bordeaux, Bordeaux, France. The experiments were carried out in accredited animal facilities following European recommendations for laboratory animal care (directive 86/609 CEE of 24/11/86). Briefly and as described in our previous study [25,26], the surgical procedure after the inhaled anesthesia was performed (Isofluorane 2% in air), comprising a skin disinfection on the lateral aspects of the femurs before a skin incision to expose the lateral aspect of the bone femurs in order to drill the bone defects (4 mm diameter and 6 mm depth) in the left and right femoral condyles. 

The holes were rinsed with physiological saline solution (0.9% NaCl (*w*/*v*)) to remove bone pieces from the defects, before the three groups of materials (Matrix-HA, Matrix-8Sr-HA, or Matrix-50Sr-HA) were injected in the defects. To prepare the Matrix-HA groups, 150 µL of sterile 0.9% NaCl was added and mixed in 1 mL syringes loaded with Matrix-HA micro beads, as described in our recent works [26]. 

After the bone defects were filled, the muscles and skin were sutured using 4.0 vicryl^®^ sutures. The animals were given IP Buprenorphin post-operatively and the following day. After 4 weeks (W4) and 8 weeks (W8), the animals were sacrificed by CO_2_ inhalation and the femurs were dissected and fixed in a solution of 10% neutral buffered formalin during 3 days for micro-CT and histological evaluation. Six independent materials were implanted per tested material and per time point.

### 4.6. Long Segmental Bone Defect in Rat and Radiation Procedure 

These experiments were performed under the supervision of an authorized laboratory member (authorization B3310023). This study was performed in an accredited animal facility (authorization A33-063-917). The rats had full, unlimited access to food and water. They were able to move without restriction. Sixty-four female Wistar-RjHan rats (12 weeks age, 250 g average weight) were obtained from a patented supplier (Janvier, Le Genest, France). Surgical procedures were performed under general anesthesia using isoflurane inhalation (Medical Sega Electronique, Lormont, France). 

The postero-lateral side of the femoral bone was approached through a 3 cm skin incision. Bone was then drilled under saline irrigation to produce a 6 mm critical bone defect. Bone defect was filled with a polymethylmetacrylate (PMMA) spacer, introduced on the left and right sides in each animal. In order to stabilize the proximal and distal femoral bone, a 5-hole titanium mini-plate (VP1313.09, Synthes, Solothurn, Switzerland) was placed, with two proximal and two distal cortical screws (1.5 mm in diameter 7 mm in length, VS102.007, Synthes, Solothurn, Switzerland). Absorbable 3/0 sutures (vicryl 4.0, Ethicon, division of Johnson & Johnson, Brussels, Belgium) were used to close the muscles and the superficial fascia. The skin was closed with Michel staples and then covered with aluminum spray (Aluspray, Vetoquinol, Lure, France). Animals were given a subcutaneous injection of a cephalosporin antibiotic (cefazoline 0.06 mg/kg) and an opioid painkiller (buprenorphine 0.05 mg/kg) during the procedure and the day after. 

Afterwards, rats were divided in two groups. In the irradiated group, the radiation delivery procedure started 3 weeks after spacer insertion. External Beam RadioTherapy (EBRT) was delivered at PRECI, an experimental veterinary radiotherapy platform (Villeneuve d’ASCQ, France.). An orthovoltage X-ray source (PANTAK, THERAPAX DXT 300 Gulmay Medical, Camberley, Surrey, UK) was used for the delivery of a 100 kV low- energy photon beam. Rats were treated individually, under isoflurane anaesthesia, in a dorsal recumbency. A 5 cm circular applicator (SSD 30cm; 3 cm efficient field width) was directed, as a single field, on the bone defect over the medial aspect of each left and right hind limb. For each field, a 3 mm sheet of lead blocked the ipsilateral aspect of the pelvis. Twenty-five daily fractions (5 days/week) of 2 Gray were administered to the implant (skin dose of 2.2 Gy) for a total dose of 50 Gray (total skin dose of 55 Gy). 

Rats were evaluated every day and weighed weekly during the EBRT procedure, as well as the period between EBRT and euthanasia. Radiation-related complications were screened: local skin inflammation, wound complication, and hair modification. Twelve weeks after the first surgery, defects were opened and PMMA spacers were removed during a second surgery. Cavities were thereafter filled with the microbeads of Matrix-HA, Matrix-8Sr-HA, or Matrix-50Sr-HA. The bone defect was closed following the same procedure than for the first step of surgery. Rats were sacrificed 12 weeks after the second surgery with a CO_2_ overdose. Samples were placed in a 4% (*w*/*v*) paraformaldehyde solution for 24 h, then kept in 0.1 M PBS pH 7.4 at 4 °C for micro-CT, histological, and immunohistochemical analysis. The experimental design with the two surgical steps is summarized on Figure 1. 

### 4.7. Micro-Computed Tomography (Micro-CT)

For the two different experimental models (femoral condyle and segmental bone defects), micro-CT imaging was performed using an Explore Locus SP x-ray microscanner device (General Electric, Milwaukee, WI). The set-up comprised a voltage of 80 kV, an intensity of 60 µA, and a time of exposure of 3000 ms to obtain a resolution of 15 µm. The ExploreMicroview^®^ software ABA 2.2 was used to reconstruct and analyze the images for the three main groups, Matrix-HA, Matrix-8Sr-HA and Matrix-50Sr-HA. The volume of interest (VOI) was determined as the volume of the initial bone defect, enabling the quantification of the new bone volume related to total volume on the samples (mineralized volume/total volume = MV/TV).

### 4.8. Histology

For both models (femoral condyle and segmental bone defects), bone explants were demineralized (Microdec^®^, Diapath, MM Brignais, France), dehydrated, and embedded in paraffin. Eight microns sections were cut and stained with Masson’s trichrome for osteoid staining, using standard protocols, and were imaged with an Eclipse 80i light microscope (Nikon, Japan). Pictures were captured using a DXM 1200 C (Nikon, Tokyo, Japan) CCD camera.

Histological slides were scanned in a Nanozoomer 2.0 HT with a fluorescence imaging module (Hamamatsu Photonics France) using an objective UPSAPO 20X NA0.75 combined with an additional lens of 1.75 X, leading to a final magnification of 35 X. The whole surface and the newly bone surface were quantified in µm^2^ using ImageJ. Six samples per condition were processed for analysis, and one section was fully imaged and analyzed per sample. The results are expressed in percentage of new bone surface to the whole surface and are shown as mean +/− standard deviation. 

### 4.9. Immunohistochemistry of Von Villebrand Factor (vWF) for Vessels Staining

Samples were retrieved and fixed in 4% (*w*/*v*) paraformaldehyde for 24h at 4 °C. Then, they were decalcified with a Microdec^®^ (MM Brignais, France) over 14 days at room temperature, dehydrated and embedded in paraffin. Sections (7–9 μm in thickness) were deparaffinized using toluene, rehydrated in decreasing concentrations of ethanol (100–50%), and finally, washed in distilled water. Antigen recovery was performed with proteinase K, diluted at 1/20 within TE buffer (50 mM Tris Base, 1mM EDTA, and pH 8.0) at room temperature for 45 min. After washing with PBS 0.1M, pH 7.4, samples were permeabilized with a Triton X100 at 0.1% (in PBS 0.1M, pH 7.4) over 10 minutes at room temperature. Then, endogenous peroxidase was quenched in 3% H_2_O_2_, in PBS, for 5 min. 

After washing with PBS 0.1M pH 7.4, slides were blocked using 2% (*w*/*v*) BSA serum in PBS 0.1M pH 7.4 for 30 min. Incubation was performed overnight with a rabbit polyclonal anti-rat vWF antibody (dilution 1:100; A0082; Dako—Agilent, Santa Clara, CA, USA) at 4 °C. After two washes with PBS 0.1M pH 7.4, the slides were incubated according to the manufacturer’s instructions (Anti-rabbit ABC kit; Vector Labs Burlingame, CA, USA), and then revealed using the Impact DAB solution (Vector Labs, Burlingame, CA, USA). Staining was stopped in distilled H_2_O.

Samples were then counterstained in Mayer’s hematoxylin and washed in running tap water for 10 min. After slides scanning, the number of vessels within the tissue was quantified using NDP view software. The whole surface and the number of vessels were quantified in mm^2^. Stained slides from two samples per condition were processed for immunostaining analysis, and three sections were fully imaged and analyzed per sample and per condition. The results are shown as averages with standard deviation per condition. 

### 4.10. Statistical Analyses

Statistical significance was evaluated with a non-parametric two-way analysis of variance (ANOVA), followed by a Tukey’s multiple comparison test, to compare all possible pairs of means independently, as provided by the GraphPad Prism Software 8.2.1 (La Jolla, CA, USA). Differences were considered significant when *p* < 0.05 (*).

## 5. Conclusions

Taken together, these results evidence that these biodegradable polysaccharide microbeads can support the regeneration of bone defects, including critical-sized bone defects, even after radiotherapy, a central clinical challenge in bone reconstruction. 

The presence of Sr within the matrix improves the degree of mineralization and the vessel density of the repaired tissue. The optimal ratio of Sr substitution for achieving the formation of vascularized mineralized tissues remains dependent on the preclinical models (size, vascular supply, time, etc.). We found that the microbeads of Matrix-8Sr-HA enhanced the vascularization of the tissue for the three preclinical models (femoral condyle, non-irradiated segmental bone model, and irradiated segmental bone model), and this opens the way for the design of future clinical applications.

## Figures and Tables

**Figure 1 ijms-24-05429-f001:**
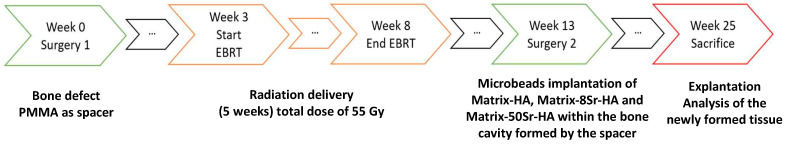
Radiation and matrices implantation procedure: the different steps of the two surgeries according to Masquelet method and the radiation delivery procedure are summarized in this scheme. EBRT: External Beam RadioTherapy.

**Figure 2 ijms-24-05429-f002:**
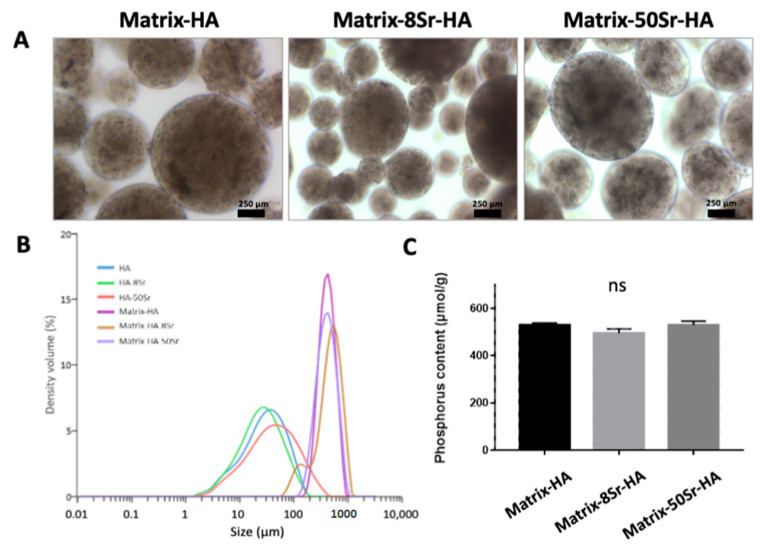
Characterization of the microbeads doped with strontium: (**A**) Morphology of composite beads prepared using hydroxyapatite doped or not with strontium. Results were obtained after immersion of freeze-dried beads in PBS 0.1M pH 7.4. Images obtained by contrast phase microscopy (objective ×10). (**B**) Size distribution of hydroxyapatite +/− strontium, and of composite beads prepared using the same concentration of polysaccharides, cross-linking agent (STMP), and hydroxyapatite, doped or not with strontium. Results after immersion of freeze-dried beads in PBS 0.1M, pH 7.4, are mean values ± SD (*n* = 6) obtained using a laser dynamic scattering particle size-measuring instrument. (**C**) Phosphorus content of cross-linked composite beads was determined using a colorimetric method. Results are presented as mean values ± SD (*n* = 4). ns: no significant differences. Results are presented as mean values ± SD (*n* = 4); ns: no significant differences.

**Figure 3 ijms-24-05429-f003:**
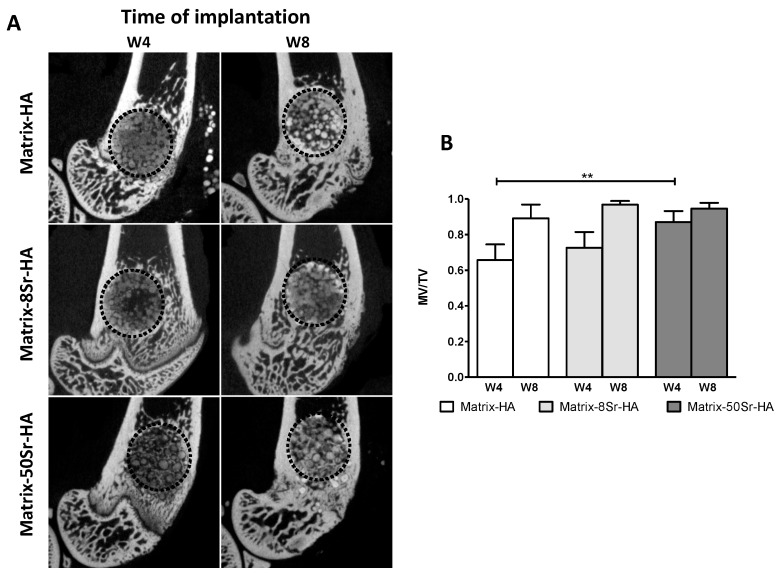
Micro-CT analysis of the tissue formed within the Matrix-HA groups, supplemented or not with strontium: (**A**) Representative Micro-CT images of these matrices 4 weeks (W4) and 8 weeks (W8) after implantation within the femoral condyle bone defect. The defect is represented with the black dote circle. (**B**) Quantification of the mineralized volume/total volume (MV/TV) 4 weeks (W4) and 8 weeks (W8) after implantation. Eight samples were evaluated for each condition (Matrix-HA, Matrix-8Sr-HA, and Matrix-50Sr-HA) at each time point (4 and 8 weeks). Results are expressed as average ± SD. The symbol ** denotes *p* < 0.01 for Matrix-50Sr-HA versus Matrix-HA at 4 weeks.

**Figure 4 ijms-24-05429-f004:**
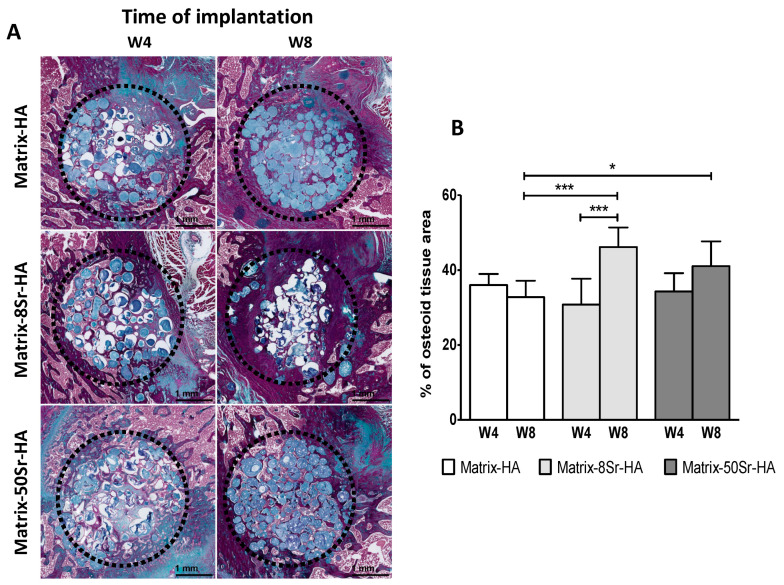
Histological analysis of the newly formed tissue the within the Matrix-HA groups supplemented or not with strontium: (**A**) Masson’s trichrome staining of the newly tissue formed within the three groups of microbeads 4 (W4) and 8 weeks (W8) after implantation within the femoral condyle bone defect. The defect is represented with the black dote circle. (**B**) Quantitative analyses of the newly bone surface in mm². Results are expressed as average ± SD per group of microbeads (Matrix-HA, Matrix-8Sr-HA, and Matrix-50Sr-HA) with time of implantation (4 and 8 weeks: W4, W8). The symbols * and *** denote *p* < 0.05 and *p* < 0.001, respectively.

**Figure 5 ijms-24-05429-f005:**
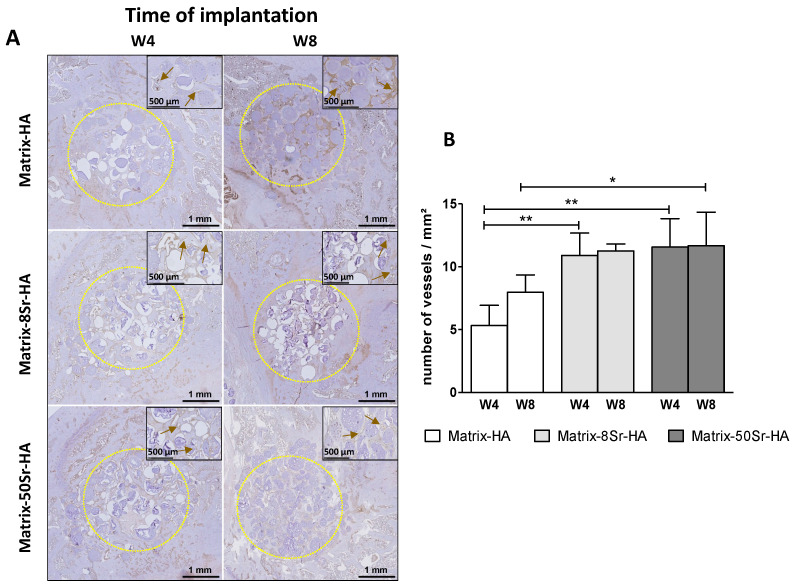
Immunohistochemistry of vWf in the newly formed tissue within the Matrix-HA groups supplemented or not with strontium: (**A**) vWF immunostaining of the newly formed tissue within the three groups of microbeads 4 (W4) and 8 weeks (W8) after implantation within the femoral condyle bone defect. The defect is represented with the doted circle. (**B**) Quantification of the number of vessels within the tissue was performed. Results are expressed in number of vessels/mm^2^ as average ± SD per group of microbeads (Matrix-HA, Matrix-8Sr-HA, and Matrix-50Sr-HA) with time of implantation (4 and 8 weeks: W4, W8). The symbols * and ** denote *p* < 0.05 and *p* < 0.01, respectively. High magnifications of images are indicated for each group and each time point. The brown arrows indicate the staining of the vessels.

**Figure 6 ijms-24-05429-f006:**
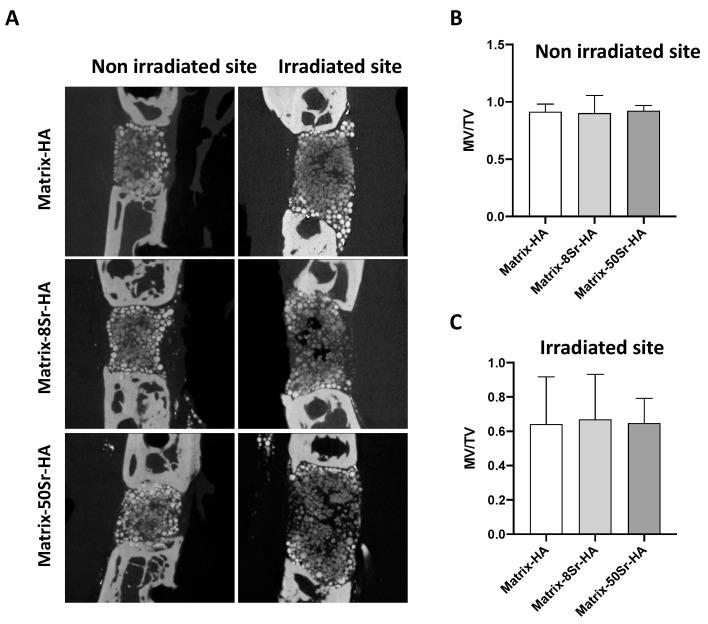
Micro-CT analysis of the tissue formed within the Matrix HA groups supplemented or not with strontium in the rat segmental bone defect models after irradiation procedure: (**A**) Representative Micro-CT images of the three groups of microbeads (Matrix-HA, Matrix-8Sr-HA, and Matrix-50Sr-HA) after 12 weeks of implantation of the microbeads within the segmental bone defects without or after irradiation procedure (non-irradiated versus irradiated site). (**B**,**C**) Quantification of the mineralized volume/total volume (MV/TV): eight samples were evaluated for each group of microbeads (Matrix-HA, Matrix-8Sr-HA, and Matrix-50Sr-HA) and for both conditions: (**B**) non-irradiated site versus (**C**) irradiated site. Results are expressed as average ± SD.

**Figure 7 ijms-24-05429-f007:**
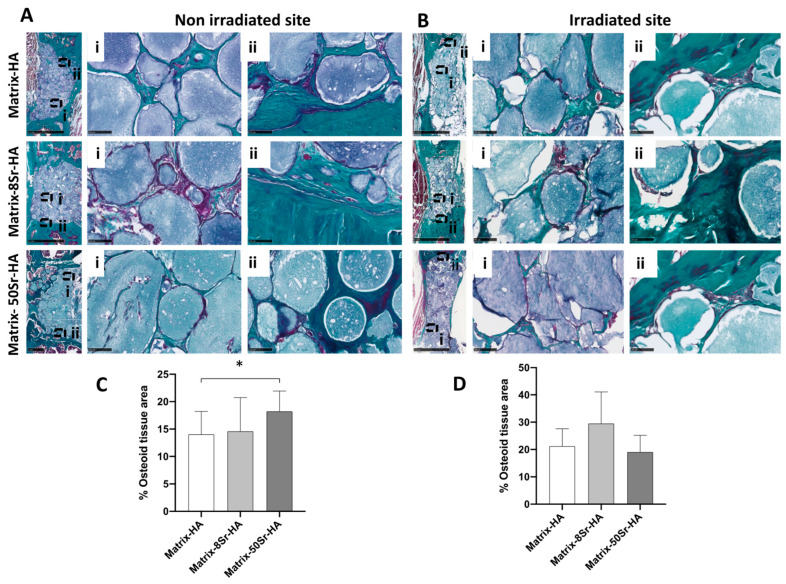
Histological analysis of the newly formed tissue the within the Matrix-HA groups supplemented or not with strontium after radiation procedure: (**A**,**B**) Masson’s trichrome staining of the newly formed tissue within the three groups of microbeads (Matrix-HA, Matrix-8Sr-HA, and Matrix-50Sr-HA) without or after irradiation procedure: (**A**) non-irradiated versus (**B**) irradiated site after 12 weeks of implantation. (**C**,**D**) Quantitative analyses of the newly bone surface in mm². Results were expressed as average ± SD per group of microbeads (Matrix-HA, Matrix-8Sr-HA, and Matrix-50Sr-HA) for both groups: (**C**) non-irradiated site versus (**D**) irradiated site. The symbol * denotes *p* < 0.05.

**Figure 8 ijms-24-05429-f008:**
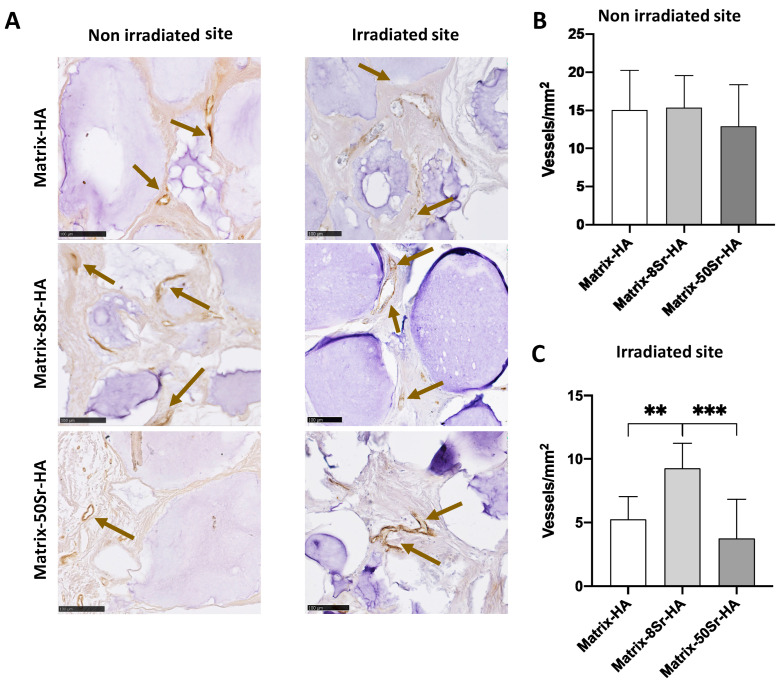
Immunohistochemistry of vWf in the newly formed tissue within the Matrix-HA groups supplemented or not with strontium after radiation procedure: (**A**) vWF immunostaining of the newly formed tissue within the three groups of matrices (Matrix-HA, Matrix-8Sr-HA, and Matrix-50Sr-HA) without or after irradiation procedure (non-irradiated versus irradiated site) after 12 weeks of implantation. Scale bar on the left and right panel: 100 μm. (**B**,**C**) Quantification of the number of vessels within the tissue was performed. Results were expressed as average ± SD per group of microbeads (Matrix-HA, Matrix-8Sr-HA, and Matrix-50Sr-HA) for both conditions: (**B**) non-irradiated site versus (**C**) irradiated site. The symbols ** and *** denote *p* < 0.01 and *p* < 0.001, respectively. The brown arrows indicate the staining of the vessels.

## Data Availability

The data that support the findings of this study are available from the corresponding author upon reasonable request.

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
