# Peer review of "Bone Regeneration in Small and Large Segmental Bone Defect Models after Radiotherapy Using Injectable Polymer-Based Biodegradable Materials Containing Strontium-Doped Hydroxyapatite Particles"

_ijms, 2023, doi:10.3390/ijms24065429_

Round 1
Reviewer 1 Report
The authors analyzed the performance of Sr-doped polysaccharide-based matrix supplemented with particles of hydroxyapatite (HA) to support both bone formation and vascularization in orthotopic site in rats, using two different models in a non-irradiated bone defect model, and in irradiated bone sites. The authors revealed that the presence of strontium within the matrix modulates the mineralization of the tissue and accelerates the kinetic of mineralization with Matrix-Sr-HA.
Some major concerns and issues should be addressed before publication.
1. The Abstract section should be rewritten in a better and clear way. The objective is missing from the abstract. The authors paid more attention on what they did. Instead, they should strengthen the introduction about their results. Moreover, some important result data should be presented in this section.
2. Many sentences must be rewritten because of ambiguity as well as lengthy. For instance, sentences in line 54-57, 61-66, … etc.
3. Please introduce the merits and demerits of both polysaccharide-based matrix, HA and Sr in the introduction section. And also state the reasons why Sr- doped were chosen in the present study.
4. In line 50, references should be written in this format [5-8].
5. The conclusion in first paragraph of the introduction ended up with the limitation in pre-vascularization. However, no info was provided about vascularization.
6. The objective in lines 94-97 also should be in the abstract.
7. Have the authors studies the effect of size distribution of microbeads? These microbeads are not uniform in their particle size distribution.
8. In most figures, more details are required in the captions as well as labeling the area of interests. For example, images with a higher magnification are required. Also, point out vWf in these images. In the caption refer the colors to their corresponding site.
9. Do the (Sr-HA) polysaccharide-based microbeads still fix in the samples during the immunohistochemistry process that involves deparaffinizing and dehydrating the samples (ethanol and toluene)? Usually these types of polymers come out during this process. Therefore OCT technique is used.
10. Are these (Sr-HA) polysaccharide-based microbeads degradable? How long do they stay in the site? Do they trigger any immune response?
Author Response
Responses to Reviewer 1.
We thank the reviewer for the constructive advices that were given and that allowed us to improve the manuscript.
- The Abstract section should be rewritten in a better and clear way. The objective is missing from the abstract. The authors paid more attention on what they did. Instead, they should strengthen the introduction about their results. Moreover, some important result data should be presented in this section.
We took into consideration the comments of the reviewer. We have proposed a revised version of the Abstract page 1 in the revised manuscript.
“Reconstruction of bones remains a challenge following tumor excision and radiotherapy. Our previous study performed using polysaccharide-based microbeads that contains hydroxyapatite evidenced osteoconductivity and osteoinductive properties. New formulations of composite microbeads containing HA particles doped with strontium (Sr) at 8 or 50% were developed to improve their biological performance, and were evaluated in ectopic sites. Here, we characterized the materials by phase contrast microscopy, laser dynamic scattering particle size-measurements and phosphorus content, before their implantation into two different preclinical bone defect models in rats: the femoral condyle and the segmental bone. After 8 weeks of implantation in the femoral condyle, histology and immunohistochemistry showed that Sr-doped matrices at 8% and 50% both stimulate bone formation and vascularization. A more complex preclinical model of irradiation procedure was then developed in rats within a critical size bone segmental defect. In the non-irradiated sites, no significant differences between the non-doped and Sr-doped microbeads were observed on the bone regeneration. Interestingly, the Sr-doped microbeads at 8% of substitution outperformed the vascularization process by increasing new vessel formation on the irradiated sites. These results showed that inclusion of strontium in the matrix stimulated vascularization in a critical size model of bone tissue regeneration after irradiation.”
- Many sentences must be rewritten because of ambiguity as well as lengthy. For instance, sentences in line 54-57, 61-66, … etc.
Following the comments, we made different changes in the text:
Among the other alternatives, the induced membrane technique surgical procedure reported in 2000 by Masquelet for long bone segmental defect reconstruction [11, 12] can be used in patients in need of bone reconstruction after cancer treatment associated with extensive loss of bone tissue.
Change to:
“Among the other alternatives, Masquelet has described the procedure of the induced membrane technique surgical procedure for long bone segmental defect reconstruction [11,12]”.
Regarding the first surgical step and the influence of External Beam RadioTherapy (EBRT) on the properties of the induced membrane in a segmental bone defect in rats, our previous works showed that induced membranes produced from polymethyl methyacrylate (PMMA) maintained histological and biochemical properties after the EBRT procedures [14]. For the second procedure, several alternatives to autografts are currently proposed including the supplementation with biological components, i.e. bone marrow suspension [11], or with bone morphogenetic proteins (BMPs) [15] known for their osteoinductive properties.
Change to (revised refrences):
Regarding the first surgical step, our previous works showed that induced membranes produced from polymethyl methyacrylate (PMMA) maintained their histological and biochemical properties after the External Beam RadioTherapy (EBRT) procedures [14]. For the second step, several alternatives to autografts are currently proposed, such as the supplementation of bone substitutes with bone marrow suspension, or with bone morphogenetic proteins (BMPs) [15,16].
- Please introduce the merits and demerits of both polysaccharide-based matrix, HA and Sr in the introduction section.
We thank the reviewer and we added some missing information in the introduction of the revised version (Page 2 line 79 to 95). The references described below are also reported in this section (reference 20 to 26 in the revised manuscript).
the polysaccharide-based microbeads that contains hydroxyapatite, used in this work, have already been extensively described by our teams for bone tissue engineering applications [20–26]. Among > 20 publications, we had selected some of the most important for this work in the reference list (see list below). Our previous publications demonstrated the potential of these natural biodegradable cell-free and growth factor-free polysaccharide-based matrix available as ready to-use sterile injectable biomaterial for craniofacial and dental applications [21, 24]. We have also demonstrated that the initial radiolucent property of these matrices at the beginning of implantation is of particular interest from a clinical point of view, as bone formation inside the grafted material can be observed using a conventional X-ray analysis compared to conventional calcium phosphate-based materials or allogenic bone substitutes.
Our previous works demonstrate that our polysaccharide-based matrix supplemented with HA [20] is osteoinductive and osteoconductive. These properties were previously evidenced in ectopic site (subcutaneously) and in orthotopic sites (femoral condyle in rats, sinus lift model in sheep) [25, 26].
Our objectives in the presented work, were to test the ability of these Sr-doped matrices to regenerate two different bone defects. Two new challenges were also defined ,i.e., an evaluation in a critical size defect of load bearing bone using a polysaccharide matrix of initial low mechanical properties, and in a segmental bone defects after irradiation procedure.
And also state the reasons why Sr- doped were chosen in the present study.
As we hypothesized, the presence of Sr within the matrix improves the degree of mineralization and the vessels density of the repaired tissue. This important is a paramount importance in clinical relevant situation such as bone reconstruction after radiotherapy, where the vascular response is largely compromised. Even if the optimal ratio of Sr substitution for achieving formation of vascularized mineralized tissues remains highly dependent on the preclinical models (size, vascular supply, time …), we have evidenced that the microbeads of Matrix-8Sr-HA enhanced vascularization of the tissue for the three preclinical models (femoral condyle, non - irradiated segmental bone model, and irradiated segmental bone model), and this open the way for the design of future clinical applications.
References cited in this revised section to support our previous works in the pullulan dextran based matrices
20. Grenier, J.; Duval, H.; Lv, P.; Barou, F.; Le Guilcher, C.; Aid, R.; David, B.; Letourneur, D. Interplay between Crosslinking and Ice Nucleation Controls the Porous Structure of Freeze-Dried Hydrogel Scaffolds. Biomaterials Advances 2022, 139, 212973, doi:10.1016/j.bioadv.2022.212973.
21 Fricain, J.C.; Schlaubitz, S.; Le Visage, C.; Arnault, I.; Derkaoui, S.M.; Siadous, R.; Catros, S.; Lalande, C.; Bareille, R.; Renard, M.; et al. A Nano-Hydroxyapatite – Pullulan/Dextran Polysaccharide Composite Macroporous Material for Bone Tissue Engineering. Biomaterials 2013, 34, 2947–2959, doi:10.1016/j.biomaterials.2013.01.049.
22 Schlaubitz, S.; Derkaoui, S.M.; Marosa, L.; Miraux, S.; Renard, M.; Catros, S.; Le Visage, C.; Letourneur, D.; Amédée, J.; Fricain, J.-C. Pullulan/Dextran/NHA Macroporous Composite Beads for Bone Repair in a Femoral Condyle Defect in Rats. PLoS ONE 2014, 9, e110251, doi:10.1371/journal.pone.0110251.
23 Ehret, C.; Aid-Launais, R.; Sagardoy, T.; Siadous, R.; Bareille, R.; Rey, S.; Pechev, S.; Etienne, L.; Kalisky, J.; de Mones, E.; et al. Strontium-Doped Hydroxyapatite Polysaccharide Materials Effect on Ectopic Bone Formation. PLoS ONE 2017, 12, e0184663, doi:10.1371/journal.pone.0184663.
24 Fricain, J.C.; Aid, R.; Lanouar, S.; Maurel, D.B.; Le Nihouannen, D.; Delmond, S.; Letourneur, D.; Amedee Vilamitjana, J.; Catros, S. In-Vitro and in-Vivo Design and Validation of an Injectable Polysaccharide-Hydroxyapatite Composite Material for Sinus Floor Augmentation. Dental Materials 2018, 34, 1024–1035, doi:10.1016/j.dental.2018.03.021.
25 Grenier, J.; Duval, H.; Barou, F.; Lv, P.; David, B.; Letourneur, D. Mechanisms of Pore Formation in Hydrogel Scaffolds Textured by Freeze-Drying. Acta Biomaterialia 2019, 94, 195–203, doi:10.1016/j.actbio.2019.05.070.
26 Maurel, D.B.; Fénelon, M.; Aid‐Launais, R.; Bidault, L.; Le Nir, A.; Renard, M.; Fricain, J.; Letourneur, D.; Amédée, J.; Catros, S. Bone Regeneration in Both Small and Large Preclinical Bone Defect Models Using an Injectable Polymer‐based Substitute Containing Hydroxyapatite and Reconstituted with Saline or Autologous Blood. J Biomed Mater Res 2021, 109, 1840–1848, doi:10.1002/jbm.a.37176.
- In line 50, references should be written in this format [5-8].
The presentation of these references has been modified [5, 6], [7, 8] is now [5-8].
- The conclusion in first paragraph of the introduction ended up with the limitation in pre-vascularization. However, no info was provided about vascularization.
As mentioned in the introduction section (lines 46 to 50 in the revised manuscript), supported by the references 7 and 8, the bone tissue engineering strategies used today, are still a long way from any application in oncology as immediate post-implantation irradiation. The main limitations are that bone tumor resection results in a large complex bone defect, and that the frequently irradiation procedure induces important vascular damages of the host site. This contributes to a decreased of osteoinductive and pro-angiogenic potential of the potential grafts used for bone reconstruction Then, the reconstruction of an avascular irradiated site (Line 61 of the revised manuscript) remains a real challenge for the clinicians. The use BMPs or angiogenic growth factors remains contraindicated in patients who have an active malignancy.
- The objective in lines 94-97 also should be in the abstract.
The objectives of this work are now clarified in the revised abstract.
- Have the authors studies the effect of size distribution of microbeads? These microbeads are not uniform in their particle size distribution.
We have extensively described the characterizations of the polysaccharide-based microbeads that contains hydroxyapatite. From our first works performed and published 10 years ago [20] on bone reconstruction using massive blocs of matrices (Discs and cylinders), we have then patented a method of production that allow the use of injectable materials for these applications. The optimization has been performed by transferring the patent and the methods to a private company that in involved in the author list.
The optimal results were obtained using the size distribution of microbeads presented here. This type of size distribution could be valuable to both manage the optimal filling of the local defects, and the progressive resorption times that should be concomitant to the new bone formation.
- In most figures, more details are required in the captions as well as labeling the area of interests. For example, images with a higher magnification are required. Also, point out vWf in these images. In the caption refer the colors to their corresponding site.
As suggested by the reviewer, we added in the Figure 5 high magnification images for vWF staining. In addition, arrows were added to point out the vWf labeling.
In Figure 7, with the bone segmental defect showing two areas, images with higher magnifications (i) and (ii) were already shown for both conditions: non irradiated site and irradiated site.
In Figure 8, high magnifications are also shown to visualize with an optimal resolution the vessels in brown around the microbeads. Brown arrows were added to point out the vWf labelling.
Our objectives here were to visualize numerous microbeads, the tissue around, and the vessels around the microbeads.
- Do the (Sr-HA) polysaccharide-based microbeads still fix in the samples during the immunohistochemistry process that involves deparaffinizing and dehydrating the samples (ethanol and toluene)? Usually these types of polymers come out during this process. Therefore OCT technique is used.
As observed in all the figures, microbeads were preserved, whatever the formulations (doped or non doped materials) or the protocols used, including the immunohistochemistry procedure.
OCT technique was not considered in this work, since deparaffinizing and rehydradation in decreasing concentrations of ethanol, followed by washing in distilled water did not alter the polysaccharide-based microbeads and gave good results in immunohistochemistry.
- Are these (Sr-HA) polysaccharide-based microbeads degradable? How long do they stay in the site? Do they trigger any immune response?
These doped polysaccharide-based microbeads are degradable. We have observed degradation of the microbeads up to 3 months of implantation in bone site. No Immune response was observed whatever the site of implantation.

Reviewer 2 Report
1.Some figure captions are missing or duplicate, as shown in Figure 2, Figure 6, 7 and 8.2.In the femoral condyle bone defect, Matrix-8Sr-HA and Matrix-50Sr-HA both stimulated the formation of vessels within the newly formed tissue (Figure 5). What is the release curve of strontium ions from Sr-HA? What is the difference between Matrix-8Sr-HA and Matrix-50Sr-HA in the release of strontium ions?
3. In critical size bone defect repair, strontium of Matrix-8Sr-HA did not promote angiogenesis in the non-irradiated group, but significantly promoted angiogenesis in the irradiated group. Please explain.
Author Response
Responses to Reviewer 2
We thank the Reviewer for these comments that allowed us to improve and modify the errors in the manuscript.
- Some figure captions are missing or duplicate, as shown in Figure 2, Figure 6, 7 and 8.
We have corrected in the revised form of the manuscript all the figure captions including Figure 2, 6, 7 and 8. Duplicate sentences or missing information about parts of the figures have been corrected for all of them.
- In the femoral condyle bone defect, Matrix-8Sr-HA and Matrix-50Sr-HA both stimulated the formation of vessels within the newly formed tissue (Figure 5). What is the release curve of strontium ions from Sr-HA? What is the difference between Matrix-8Sr-HA and Matrix-50Sr-HA in the release of strontium ions?
We thank the reviewer for this comment. We added some parts in the revised version (Page 11) and the following references described below are now included in the réferences sections (References 31 to 35 in the revised manuscript)
The analysis of the Sr2+ release from these microbeads is important to confirm its biological effect in vitro but especially in vivo. In our work, although the in vitro Sr2+ release from these Matrix-8Sr-HA and Matrix-50Sr-HA were not determined, Sr 2+ release and its biological effect on osteoblastic differentiation of human mesenchymal stem cells were previously demonstrated (Ehret et al, Plos One 2017 Ref 23 in the revised manuscript.doi.org/10.1371/journal.pone.0184663). In this paper, we have also evidenced that Matrix-8Sr-HA and Matrix-50Sr-HA both activated the expression of one late osteoblastic marker involved in the mineralization process i.e. osteopontin compared to Matrix-HA devoid of strontium. In addition, only the Sr-doped matrices impacted in vivo osteoid tissue and blood vessels formation. Nevertheless, several in vitro studies reported the Sr 2+ release from Sr-doped biomaterials using a similar substitution ratio as our Matrix-8Sr-HA.
- For instance, Landi et al. (Landi E, Tampieri A, Celotti G, Sprio S, Sandri M et al. (2007) Sr- substituted hydroxyapatites for osteoporotic bone replacement. Acta Biomater 3: 961-969. DOI: 1016/j.actbio.2007.05.006.) synthesized a Sr- substituted HA with 8.7 wt% Sr content and measured the in vitro Sr2+ ion release by immersing 3 g of HA granules in 50 ml of Hanks’ balanced salt solution with 6.060 ppm Sr2+ (6.060 mg/L). After 24 h, the Sr concentration decreased to only 0.602 ppm in the immersion liquid. (Reference 32 in the revised manuscript)
- Panzavolta et al. (Panzavolta S, Torricelli P, Sturba L, Bracci B, Giardino R et al. (2008) Setting properties and in vitro bioactivity of strontium-enriched gelatin- calcium phosphate bone cements. J Biomed Mater Res A 84: 965-972 DOI: 1002/jbm.a.31412 ) described an α-TCP-based cement/gelatin composite with 5 mol % Sr. They observed low doses of Sr2+ released from bone cement soaked in physiological solution, but hypothesized this was sufficient to influence cellular response. (Reference 33 in the revised manuscript, also supported by the reference 34: Zhao et al. doi.org/10.1371/journal.pone.0069339).
- More recently, Chen L et al. (Chen L, Mazeh H, Guardia A, Song W, Begeman P, Markel DC, Ren W. Sustained release of strontium (Sr2+) from polycaprolactone/poly (d,l-lactide-co-glycolide)-polyvinyl alcohol coaxial nanofibers enhances osteoblastic differentiation. Journal of Biomaterials Applications.2019;34:533-545. org/10.1177/0885328219858736) studied the release of Sr2+ from polycaprolactone/poly (D,L-lactide-co-glycolide)-polyvinyl alcohol coaxial nanofibers. They observed a sustained Sr2+ release from the PCL/PLGA-PVA coaxial nanofibers for over two months. (Reference 31 in the revised manuscript).
A deeper analysis was recently published by Mocanu et al. (Mocanu A, Cadar O, Frangopol PT, Petean I, Tomoaia G, Paltinean G-A, Racz CP, Horovitz O, Tomoaia-Cotisel M. 2021 Ion release from hydroxyapatite and substituted hydroxyapatites in different immersion liquids: in vitro experiments and theoretical modelling study. R. Soc. 2021,Open Sci. 8, 1. doi.org/10.1098/rsos.201785.) on nano-hydroxyapatite substituted with 5 % Sr2+. The release was performed both in water and in simulated body fluid, in static and simulated dynamic conditions, using inductively coupled plasma optical emission spectrometry. The results revealed a steady release of Sr2+ ions from HAP-Sr nanomaterials in water and in SBF over 90 days. The results indicated a mechanism principally based on diffusion and dissolution, with possible contribution of ion exchange. The in vitro Sr2+release in immersion liquids (static and dynamic regime) was found within the therapeutic window of Sr concentrations of 2–45 ppm, and expected to promote osteogenesis and in vivo, bone regeneration. (Reference 35 in the revised manuscript).
Altogether, these in vitro release studies supported the concept a prolonged supply of essential ions of biological importance, indispensable for osteoblast activity and thus can contribute to the formation and development of healthy new bone tissue and bone regeneration.
However, in vitro models are highly dependent of the immersing media (water vs simulated body fluid; flow; pH) and did not take into account a possible exchangeable pool of strontium linked to proteins and no cellular event is present in vitro. It is likely that bone remodeling and the in vivo environment accelerate the Sr2+ release by mechanisms such as biomaterial resorption by cells such osteoclasts, and numerous enzymatic activities found in the injured area. Future in vivo studied are thus required to carefully decipher the mechanisms of dissolution of our Sr-HA biomaterials.
- In critical size bone defect repair, strontium of Matrix-8Sr-HA did not promote angiogenesis in the non-irradiated group, but significantly promoted angiogenesis in the irradiated group. Please explain.
We thank the reviewer for this comment. W added some parts in the Discussion on the revised manuscript (Page 13).
As observed for the femoral condyle bone defect model, Matrix-8Sr-HA this matrix seems to be optimal to promote the formation of osteoid tissue and vessel formation after 8 weeks of implantation (Figure 4B and Figure 5B). In the segmental bone defect model in rats, in absence of irradiation histological analyses (Figure 7) showed significant differences in the % of osteoid tissue area, only between Matrix-HA and Matrix-50Sr-HA groups. In the irradiated group, we only observed a trend of increasing osteoid tissue area using Matrix-8Sr-HA. In future studies, we will have to increase the number of samples to see if it is significant.
Regarding the vascularization (Figure 8), we noticed that the number of vessels/mm2 remains lower in the irradiated sites compared to non-irradiated site, whatever the group of microbeads implanted. As expected, this result confirms that the radiotherapy compromises the vascularization of the tissue and thereafter optimal bone reconstruction. Quantitative analysis of the neo-vascularization in the irradiated sites evidenced a positive effect of Matrix-8Sr-HA compared to the microbeads of Matrix-50Sr-HA and to non-doped microbeads. The very low level of vascularization of the irradiated sites compared to the non-irradiated sites could explain the impact of the Matrix-8Sr-HA only on the irradiated sites and not on the non-irradiated sites where the number of vessels is around three times higher and for which the effect could be less visible. As observed here for the femoral condyle bone defect model, Sr substitution and mainly 8Sr-HA, can modulate the vascular damage induced by radiotherapy. We thereafter complete the discussion p13, line 445-451.

Reviewer 3 Report
The manuscript entitled “Bone regeneration in small and large segmental bone defect models after radiotherapy using injectable polymer-based bio-degradable materials containing strontium-doped hydroxyapatite particles” is very interesting and in the field of research that is rapidly progressing. Below authors can find suggestions and comments that can improve the quality of the submitted study. I hope the authors will find them useful.
Overall, the abstract and introduction are very well written.
On page 2 (line 50) the “[5, 6], [7, 8]” should be written as “[5-8]”.
In the introduction, authors could give an overview of recent studies focused on substituted calcium phosphates in combination with polymers which proved the increased bone regenerative response compared to materials containing non-substituted calcium phosphates. This can further highlight the importance of this study. (e.g. https://doi.org/10.1016/j.cej.2019.01.015 , https://doi.org/10.1016/j.carbpol.2021.118883 , https://doi.org/10.1016/j.colsurfb.2017.12.048).
Authors should revise the whole manuscript regarding written formulas where numbers should be subscripted, and the charge of elements superscripted.
The materials and Methods section is written properly so other researchers can repeat the experiments.
Why did the authors select 8 and 50 % of substitution? Is this molar, atomic, or weight % of substitution? After checking the authors' previous studies on this system (reference 21) it seems that an additional inorganic phase is obtained along the HA phase (XRD analysis, peak around 29 °). The authors should check if the precursor of strontium ions (Sr(NO3)2) stayed unreacted. The ICDD card of HA does not indicate a peak at this position. Due to the similar ion radius of Ca2+ and Sr2+ ions, Sr2+ is one of the easier elements to substitute in HA. However, numerous studies investigated the limit of this substitution, which highly depends on the preparation method. Authors should identify additional inorganic phase precipitated. EDS analysis can further check the distribution of the strontium ions in HA powder. The authors should check how much strontium was substituted in the system. Sometimes, initially added and obtained substitutions differ.
Author Response
Responses to Reviewer 3
We thank the Reviewer for these comments and suggestions that allowed us to improve the introduction and Materials and methods sections.
The manuscript entitled “Bone regeneration in small and large segmental bone defect models after radiotherapy using injectable polymer-based bio-degradable materials containing strontium-doped hydroxyapatite particles” is very interesting and in the field of research that is rapidly progressing. Below authors can find suggestions and comments that can improve the quality of the submitted study. I hope the authors will find them useful.
Overall, the abstract and introduction are very well written.
On page 2 (line 50) the “[5, 6], [7, 8]” should be written as “[5-8]”.
The presentation of these references has been modified [5, 6], [7, 8] is now [5-8].
- In the introduction, authors could give an overview of recent studies focused on substituted calcium phosphates in combination with polymers which proved the increased bone regenerative response compared to materials containing non-substituted calcium phosphates. This can further highlight the importance of this study. (e.g. https://doi.org/10.1016/j.cej.2019.01.015, https://doi.org/10.1016/j.carbpol.2021.118883 , https://doi.org/10.1016/j.colsurfb.2017.12.048).
As suggested by the reviewer, we have modified the introduction with recent relevant studies performed on composites polymers (Page 11, line 103 – 114 in the revised manuscript). The references were also added in the revised manuscript (Reference 27, 28 and 29 in the revised manuscript).
- Authors should revise the whole manuscript regarding written formulas where numbers should be subscripted, and the charge of elements superscripted.
We thank the reviewer for this remark. We revised the whole manuscript, including in the Material and Methods section with numbers subscripted, and the charge of elements superscripted.
- The materials and Methods section is written properly so other researchers can repeat the experiments.
- Why did the authors select 8 and 50 % of substitution? Is this molar, atomic, or weight % of substitution? After checking the authors' previous studies on this system (reference 21) it seems that an additional inorganic phase is obtained along the HA phase (XRD analysis, peak around 29 °). The authors should check if the precursor of strontium ions (Sr(NO3)2) stayed unreacted. The ICDD card of HA does not indicate a peak at this position. Due to the similar ion radius of Ca2+ and Sr2+ ions, Sr2+ is one of the easier elements to substitute in HA. However, numerous studies investigated the limit of this substitution, which highly depends on the preparation method. Authors should identify additional inorganic phase precipitated. EDS analysis can further check the distribution of the strontium ions in HA powder. The authors should check how much strontium was substituted in the system. Sometimes, initially added and obtained substitutions differ.
As described in our previous work (Erhet et al Plos One 2017, Ref 23 in the revised manuscript), HA particles were synthesized by wet chemical precipitation according to Catros et al (Ref 44 in the revised manuscript) by using 1.08 M Ca(NO3)2, 4H2O solution and a solution of 0.65 M (NH4)2HPO4 solution, heated at 90°C. The pH of solution was adjusted to 10 with NH4OH, added dropwise under stirring. The precipitate was maintained for 5 h at 90°C under stirring, and then washed 4 times with CO2-free distilled water. For strontium substitution, the doped Sr-HA was obtained following the same procedure by adding Sr2+ions into the Ca2+ solution, before adjusting the pH to 10 with NH4OH. The appropriate amounts of Sr (NO3)2 and Ca (NO3)2·4H2O were dissolved in order to obtain 8% or 50% (molar ratio) of Ca2+ substituted by Sr2+, 8Sr-HA and 50Sr-HA, respectively.
These two ratios of substitution were selected according to the literature for their impact on osteogenesis and bone formation (these References are now included in the section: references 31, 32, 33, 34, in the revised manuscript:
- Landi et al. (Landi E, Tampieri A, Celotti G, Sprio S, Sandri M et al. (2007) Sr- substituted hydroxyapatites for osteoporotic bone replacement. Acta Biomater 3: 961-969. DOI: 1016/j.actbio.2007.05.006.) (Reference 32 in the revised manuscript).
- Panzavolta et al. (Panzavolta S, Torricelli P, Sturba L, Bracci B, Giardino R et al. (2008) Setting properties and in vitro bioactivity of strontium-enriched gelatin- calcium phosphate bone cements. J Biomed Mater Res A 84: 965-972. DOI: 1002/jbm.a.31412 ) (Reference 33 in the revised manuscript, also supported by the reference 34: Zhao et al. doi: 10.1371/journal.pone.0069339).
- Chen L et al. (Chen L, Mazeh H, Guardia A, Song W, Begeman P, Markel DC, Ren W. Sustained release of strontium (Sr2+) from polycaprolactone/poly (d,l-lactide-co-glycolide)-polyvinyl alcohol coaxial nanofibers enhances osteoblastic differentiation. Journal of Biomaterials Applications.2019;34:533-545.org/10.1177/0885328219858736). (Reference 31 in the revised manuscript).
In our previous work (Erhet et al Plos One 2017, Ref 23 in the revised manuscript), we did not perform the EDS analysis for the distribution of the strontium ions in the HA powder. Only XRD patterns and FTIR spectra of non-substituted and Sr-substituted HA were done. These Sr-doped hydroxyapaptite particles were also characterized by Inductively coupled plasma optical emission spectrometry (ICP-OES) for the determination of the Sr2+ and Ca2+content. With increasing Sr-substitution in the HA particles (8% and 50% of substitution), ICP-OES showed an increasing amount of Sr in the HA samples from 2 ± 1 μg / mg of powder (8Sr-HA) to 9.7 ± 3.8 μg / mg of powder (50Sr-HA).
Concerning the mode of substitution, Kołodziejska et al (The Influence of Strontium on Bone Tissue Metabolism and Its Application in Osteoporosis Treatment, Int J Mol Sci. 2021 Jun; 22(12): 6564. doi: 10.3390/ijms22126564) have described that strontium ions can be incorporated into the structure of calcium phosphate apatites in two ways. The first (is adsorption on the surface of the mineral. The second takes advantage of the chemical similarity between Sr and Ca. Sr ions easily incorporate into the crystalline lattice structure, replacing calcium.
In our previous work (Erhet et al Plos One 2017, Ref 23 in the revised manuscript), we did not investigate the limit of our substitution method. Nevertheless ICP-OES analysis allowed us to quantify both the calcium and strontium content in the hydroxyapatite samples doped with the two Sr substitution ratios. These results are shown in the Table 4 of the publication by Ehret et al (Erhet et al Plos One 2017, Ref 23 in the revised manuscript) and confirm the Sr substitution. Finally, the biological effects of strontium substitution were demonstrated in vitro on the differentiation of human mesenchymal stem cells to the osteogenic lineage and in vivo on the angiogenesis after subcutaneous implantation of the matrices.

Round 2
Reviewer 3 Report
-